# How Do We Monitor Oxygenation during the Management of PPHN? Alveolar, Arterial, Mixed Venous Oxygen Tension or Peripheral Saturation?

**DOI:** 10.3390/children7100180

**Published:** 2020-10-13

**Authors:** Praveen Chandrasekharan, Munmun Rawat, Satyan Lakshminrusimha

**Affiliations:** 1Division of Neonatology, Department of Pediatrics, University at Buffalo, Buffalo, NY 14260, USA; munmun.rawat@gmail.com; 2Division of Neonatology, Department of Pediatrics, University of California Davis, Davis, CA 95616, USA; slakshmi@ucdavis.edu

**Keywords:** oxygenation, PPHN, oxygen tension

## Abstract

Oxygen is a pulmonary vasodilator and plays an important role in mediating circulatory transition from fetal to postnatal period. Oxygen tension (PO_2_) in the alveolus (PAO_2_) and pulmonary artery (PaO_2_) are the main factors that influence hypoxic pulmonary vasoconstriction (HPV). Inability to achieve adequate pulmonary vasodilation at birth leads to persistent pulmonary hypertension of the newborn (PPHN). Supplemental oxygen therapy is the mainstay of PPHN management. However, optimal monitoring and targeting of oxygenation to achieve low pulmonary vascular resistance (PVR) and optimizing oxygen delivery to vital organs remains unknown. Noninvasive pulse oximetry measures peripheral saturations (SpO_2_) and a target range of 91–95% are recommended during acute PPHN management. However, for a given SpO_2_, there is wide variability in arterial PaO_2_, especially with variations in hemoglobin type (HbF or HbA due to transfusions), pH and body temperature. This review evaluates the role of alveolar, preductal, postductal, mixed venous PO_2_, and SpO_2_ in the management of PPHN. Translational and clinical studies suggest maintaining a PaO_2_ of 50–80 mmHg decreases PVR and augments pulmonary vasodilator management. Nevertheless, there are no randomized clinical trials evaluating outcomes in PPHN targeting SpO_2_ or PO_2_. Also, most critically ill patients have umbilical arterial catheters and postductal PaO_2_ may not be an accurate assessment of oxygen delivery to vital organs or factors influencing HPV. The mixed venous oxygen tension from umbilical venous catheter blood gas may assess pulmonary arterial PO_2_ and potentially predict HPV. It is crucial to conduct randomized controlled studies with different PO_2_/SpO_2_ target ranges for the management of PPHN and compare outcomes.

## 1. Introduction

Persistent pulmonary hypertension of the newborn (PPHN) occurs when there is impaired pulmonary vascular transition during birth due to disruption of pulmonary vasodilator mechanisms. Impaired transition from fetal to neonatal circulation leads to elevated pulmonary vascular resistance (PVR), right-to-left or bidirectional shunts at patent foramen ovale (PFO) and/or patent ductus arteriosus (PDA) leading to hypoxemia [1]. Both term and preterm neonates are at risk for PPHN [2,3,4,5]. The course of preterm neonates in the neonatal intensive care unit (NICU) is further complicated by the development of bronchopulmonary dysplasia (BPD) and can potentially be associated with pulmonary hypertension (PHT) [6,7,8]. The incidence of PPHN in neonates are often underestimated but is thought to be around 1.8 to 2/1000 live births [1,5,9]. In infants with PPHN, several studies report poor long-term neurodevelopment outcomes and higher early mortality rates despite pulmonary vasodilator therapies [1,10,11]. At birth, with the initiation of spontaneous breathing or with positive pressure ventilation (PPV) and with adequate lung inflation, the pulmonary blood flow (PBF) increases by 8 to 10-fold along with a decrease in PVR. As the fetus grows in a relatively hypoxemic environment, increase in oxygen tension seems to play a significant role in decreasing PVR at birth [12]. Optimal oxygenation is necessary to meet tissue demand, especially in vital organs such as brain and heart and to prevent hypoxic pulmonary vasoconstriction (HPV). This review discusses the role of oxygen tension (PO_2_) and pulse oximetry (SpO_2_) during the management of PPHN. With lack of clinical evidence on optimal oxygenation in PPHN, this manuscript reviews data from both term and preterm translational models associated with high PVR in the perinatal period.

## 2. Discussion

Understanding the relationship of fetal oxygenation, PVR, pulmonary blood flow (PBF) in both the fetus and newborn is critical to managing a neonate with PPHN.

### 2.1. Relation of PO_2_ and Fetal PVR

The fetus develops with placenta serving as an organ of gas exchange. The highest PO_2_ within the fetal circulation is approximately 32–35 mmHg in the umbilical vein. There is a further decrease in PO_2_ to 25–28 mmHg in the ascending aorta supplying the developing brain and myocardium [13,14,15]. The placenta protects the fetus from maternal hyperoxia and hypoxia [14,15]. As observed in translational studies, during maternal hyperoxia/hypoxia, the distribution of blood in the placenta, channeling of blood to and from the fetal liver by ductus venosus, and alteration of PVR (increase/decrease) avoids excessive fluctuations in fetal PO_2_ [16]. The relationship of fetal PVR to PO_2_ is dependent on the gestational age (GA) of the fetus [17]. By term gestation, PVR decreases dramatically in response to increased fetal PO_2_ [18].

### 2.2. Gestational Age, Fetal PVR, and PO_2_

The changes in PVR to PO_2_ in relation to GA were studied in fetal ovine model [19]. Ovine fetuses at different GA of 0.6 (103–104/term ~150 days), 0.74 (112–119 days), 0.80 (121–130 days), and 0.90 (132–138 days) were exposed to hypoxia or hyperoxia by adjusting the oxygen exposure to the ewe. Pulmonary vasoconstriction and vasodilation were observed at term gestation when exposed to low and high PO_2_. Hypoxia and hyperoxia did not have a significant effect on PVR at 0.6 and 0.74 gestation. In humans, maternal hyperoxia did not alter fetal PBF at 20–26 weeks GA but increased PBF and reduced atrial and ductal shunting at 31 to 36 weeks [20]. Extrapolating from these findings, the pulmonary vascular response to PO_2_ seems to increase with advancing gestational age.

### 2.3. Effect of PO_2_ on PVR at Birth

During normal transition, spontaneous breath initiated by the newborn infant ventilates the lung and increases alveolar oxygen tension, which increases PBF reducing PVR, successfully switching from fetal to neonatal circulation [21]. Multiple factors such as mode of delivery (vaginal delivery results in more rapid reduction in PVR compared to an elective cesarean section), maturity, antenatal glucocorticoids, temperature (hypothermia increases PVR), mode of cord clamping and asphyxia (related to higher PVR), could affect transition at birth [22]. These factors affect the balance between the vasoconstrictors (endothelin-1, thromboxane), and vasodilators (prostacyclin and endothelium derived nitric oxide), which exert their effects on the pulmonary artery smooth muscle cells (PASMC). In addition to these factors, oxygen (O_2_) seems to play a greater role in the regulation of PVR by having a direct effect on PASMC. Oxygen stimulates the increased production of pulmonary endothelial nitric oxide (NO), which is a potent pulmonary vasodilator birth [22].

### 2.4. Oxygen and Hypoxic Pulmonary Vasoconstriction (HPV)

A fundamental difference between pulmonary blood vessels and systemic vessels is their ability to constrict in response to hypoxia [23]. Regional HPV diverts blood away from underventilated alveoli and promotes ventilation-perfusion (V/Q) matching. HPV is mediated by PO_2_ surrounding the precapillary pulmonary arteriole (Figure 1) and is influenced both by mixed venous (pulmonary arterial) PVO_2_ and alveolar PAO_2_ [24]. The stimulus for HPV is dictated by the equation P(stimulus)O_2_ = PVO_2_^0.375^ × PAO_2_^0.626^. Based on this equation, it is clear that alveolar PAO_2_ is the predominant factor determining the severity of HPV. Acidosis exacerbates HPV in neonatal animal models [15].

### 2.5. Supplemental Oxygen and PO_2_ during the Transition

Oxygen supplementation is the most common resuscitative measure for newborns in the delivery room. American Academy of Pediatrics Neonatal Resuscitation Program (AAP NRP) recommends that supplemental O_2_ be initiated at concentrations of 21% O_2_ in term and 21–30% O_2_ in preterm neonates and to titrate based on prespecified preductal saturations [25]. Given the ease and universal use of pulse oximetry, preductal oxygen saturations (SpO_2_) could be the most efficient way of targeting oxygenation in the delivery room and the neonatal intensive care unit (NICU). However, for a given saturation range, the achieved PO_2_ could vary widely and the extremes of SpO_2_ have low accuracy [26]. In a newborn requiring resuscitation, hypoperfusion could also decrease the accuracy of SpO_2_.

### 2.6. Oxygen Tension in Spontaneous Air Breathing Infants

The concept of normoxemia in a transitioning newborn is not well defined. A healthy newborn, who transitioned from placenta to lungs as an organ of gas exchange, sees a rise in arterial oxygen tension (PaO_2_) by 30–40 mmHg from fetal values. The increases in alveolar PAO_2_ and arterial PaO_2_ along with ventilation play an important role in decreasing PVR and increasing PBF. In a study of near term gestation lambs (comparable to human term neonates), the use of 21% O_2_ lead to PaO_2_ values of 50–60 mmHg [12]. The observed decrease in PVR from fetal life occurred at a PaO_2_ of 52.5 ± 1.7 mmHg, also known as the change point [12]. The change point is the PaO_2_ value at which there is a change in the slope of PVR-PaO_2_ scatter plot. In human neonates, who were previously on respiratory support and were weaned to room air and breathing spontaneously, 176 samples of arterial blood gas were analyzed [27]. The analysis showed that 80% of the PaO_2_ values were between 40–80 mmHg and the average PaO_2_ was 64 mmHg. In a study involving 10 neonates, the average PaO_2_ was 77 ± 4.5 mmHg in quiet asleep compared to a PaO_2_ of 70 ± 6.5 mmHg in active sleep [28]. In this study, the quiet sleep was defined as no eye movement or activity, while active sleep had frequent small movements, and the site of blood draw (pre vs. post ductal) is not known [28]. A recent study, defined normoxemia in neonates with a PaO_2_ range of 50–80 mmHg [29]. Based on these observations, the PaO_2_ in normal neonates mostly ranges between 50–80 mmHg. Achieving a similar preductal PaO_2_ value during the management of PPHN neonates appears prudent.

### 2.7. Oxygen Tension and PPHN

In a neonate with PPHN secondary to failed transition or underlying lung pathology, the elevated pulmonary pressures often lead to shunting of blood from pulmonary to the systemic circulation, leading to profound and labile hypoxemia despite PPV and supplemental oxygen. Adequate oxygenation to achieve preductal PaO_2_ in the 50–80 mmHg range (similar to normal neonates) remains the cornerstone of PPHN management in both term and preterm neonates.

### 2.8. Alveolar PAO_2_ and its Effect on PPHN

The alveolar PO_2_ (PAO_2_), which can be calculated using the inspired O_2_ concentration and arterial oxygenation and carbon dioxide tension, is a major determinant of HPV [30]. In the presence of parenchymal lung injury and or immature lungs, alveolar hypoxia could exacerbate pulmonary vasoconstriction. Studies done four decades ago, comparing newborn and adult ovine models, have shown that alveolar hypoxia leads to more significant HPV leading to redistribution of pulmonary circulation away from hypoxic lung in newborn lambs compared to adult sheep (Figure 2). 

In adult sheep, alveolar PAO_2_ had to drop to 19 mmHg to observe significant redistribution of blood flow away from the hypoxic lung. In sharp contrast, reduction of alveolar PAO_2_ below 360 mmHg (equivalent to ventilation with 50% oxygen), lead to HPV and redistribution of blood away from the hypoxic lung (Figure 2).

However, HPV occurred only with severe alveolar hypoxia in adult lambs exacerbating ventilation-perfusion mismatch [31]. These results suggest that HPV in response to alveolar hypoxia is more prominent and severe, and occurs at higher PAO_2_ levels during the neonatal period compared to adult life.

Table 1 shows the effect of inspired oxygen concentration on preductal arterial (PaO_2_), PVR and alveolar PAO_2_ (calculated using PaCO_2_), in control term gestation lambs and lambs with PPHN induced by antenatal ductal ligation (a model of primary PPHN without any lung disease). [12] In control lambs without lung disease or PPHN, low PVR is achieved by ventilation with 21% oxygen with alveolar PAO_2_ in the 90–100 mmHg range.

However, in lambs with PPHN, use of 21% oxygen leads to normal PAO_2_ but low PaO_2_ compared to non-PPHN controls, and is associated with high PVR. Increasing calculated PAO_2_ by increasing inspired oxygen led to a decrease in PVR (Table 1) [12]. In this model of primary PPHN, inspired oxygen concentration of 50% was needed to elicit maximal decrease in PVR. Although alveolar hypoxia worsens PPHN, hyperoxia (PAO_2_ > 300 mmHg) did not have any additional vasodilator effect. Pulmonary vasodilator response to hyperoxia may not be sustained and could blunt the vasodilator response to inhaled nitric oxide (iNO) [32]. A concern with alveolar PAO_2_ is that it is calculated using mathematical equations and in heterogenous lung disease, different areas of the lung may have different PAO_2_ values.

In an acute meconium aspiration model of ovine PPHN, SpO_2_ range of 90–94% was associated with an increase in PaO_2_ alone without an increase in PAO_2_ and resulted in a marginal improvement in PVR and PBF. At SpO_2_ of 93–97% an increase in both PaO_2_ and PAO_2_ were necessary to achieve optimal decrease in PVR (Figure 3). With the high prevalence of PHT among extremely preterm infants, it is critical to understand the role of PAO_2_ in regulating PVR [33]. Kinsella et al. evaluated the pulmonary vasodilator effect of iNO, oxygen and lung distension on developing pulmonary circulation [34]. In preterm lambs at 115 days GA (0.78 of term), pulmonary vasodilation was more prominent with rhythmic lung distension compared to supplemental oxygen or iNO [34]. In near term lambs, 100% oxygen and iNO led to profound pulmonary vasodilation [34]. We speculate that in the setting of extreme prematurity and respiratory distress syndrome in the absence of classic PPHN physiology, iNO and alveolar hyperoxia may not be effective in decreasing PVR compared to term neonates.

### 2.9. Arterial Oxygen Tension and Its Effect on PPHN

Preductal arterial oxygenation (PaO_2_) is typically used to assess the severity (based on oxygenation index), management, and evaluate response to therapy in PPHN [35]. In neonates with PPHN treated with adequate ventilation and supplemental oxygen, a preductal PaO_2_ of < 40 mmHg reflects hypoxemia. Secondary to right-to-left shunting across the ductus, there may be a pre and post ductal difference in SpO_2_ and PaO_2_. No clinical studies to date have studied the effect of maintaining various levels of PaO_2_ in the management of PPHN. As mentioned previously, a preductal PaO_2_ of > 52.5 ± 1.7 mmHg decreased PVR in ovine models without PPHN, while a PaO_2_ of > 59.6 ± 15.3 mmHg was required to decrease PVR in a PPHN model [12]. In a preterm RDS model, a PaO_2_ of > 58 mmHg was required to decrease in PVR [17]. Similarly, the change point for preductal PaO_2_ was 45 ± 0.1 mmHg in a model of asphyxia with meconium aspiration syndrome.

Recently, we have shown that in a model of asphyxia and meconium aspiration with PPHN during the first 6 h post-resuscitation, targeting a preductal SpO_2_ of 95–99% and achieving 93–97% SpO_2_ with a corresponding PaO_2_ of 58 ± 19 mmHg was associated with the lowest PVR (0.55 ± 0.15 mmHg/mL/kg/min), with an inspired oxygen requirement of 50 ± 21% [36]. In this study, the SpO_2_ range of 90–94% had a similar PaO_2_ (56 ± 11 mmHg), but the corresponding PVR was much higher with a significantly lower inspired oxygen requirement (29 ± 17%) [36]. These results shown in Figure 3 outline the importance of alveolar PAO_2_ in addition to arterial PaO_2_ in mediating lower PVR. While the PaO_2_ in the 90–94% SpO_2_ group and 93–97% SpO_2_ group were identical, we speculate that the difference in FiO_2_ contributed to the marked decrease in PVR in the 93–97% group (Figure 3).

With the available data, preductal PaO_2_ has a high utility in neonates with PPHN as it dictates the amount of oxygen delivered to the cerebral and coronary circulation. In summary, targeting preductal arterial oxygenation in the clinically accepted range of normoxemia (50–80 mmHg), could help in managing PPHN by optimizing oxygen delivery, but is not the only factor determining PVR. Providing adequate FiO_2_ to maintain optimal alveolar PAO_2_ is also important in mediating pulmonary vasodilation. Avoiding extremely high PaO_2_ (>100 mmHg) and PAO_2_ (>300 mmHg) may potentially facilitate response to other pulmonary vasodilators, reducing high cumulative oxygen exposure and toxicity [12,30,32]. The optimal PaO_2_ range during acute phase of PPHN warrants further clinical trials focusing on both short-term and long-term outcomes.

## 3. Post-Ductal PaO_2_ (PDPaO_2_) and PPHN

Given the easy arterial access path in neonates, a high umbilical arterial catheter (UAC), with its tip in the thoracic portion of the descending aorta, is the most commonly placed arterial access in the NICU [37]. When blood gas is obtained from the UAC, it usually reflects post-ductal (PDPaO_2_) unless the ductus arteriosus is closed (Figure 1) [35]. Labile hypoxemia, in the presence of shunting from pulmonary to systemic circulation, could present with pre and post ductal PaO_2_ gradient of 10–20 mmHg, which often goes in hand with pre and post ductal SpO_2_ difference and is characteristic of PPHN. In the absence of shunting, a PDPaO_2_ could reflect preductal PaO_2_. In a clinical trial comparing the blood gas to saturation values, out of 800 arterial blood gas samples collected, 88% of the samples were postductal, which reflects the extensive use of PDPaO_2_ to evaluate and guide therapy when supplemental oxygen is needed [27]. Using preductal SpO_2_ to adjust FiO_2_ is preferred to the use of PDPaO_2_ during management of PPHN. However, absence of a preductal to postductal oxygenation gradient does not rule out bidirectional ductal shunting and low PBF [35].

### 3.1. Mixed Venous Oxygen Tension (PvO_2_) and Management of PPHN

The mixed venous PO_2_ typically refers to oxygen tension in the pulmonary artery [38]. Since umbilical venous catheters (UVC) are commonly placed in the NICU, the blood gas obtained from a UVC is used as a proxy for mixed venous gas [39]. The PVO_2_ could assess the tissue oxygenation in neonates. 

In a clinical study involving 22 neonates with respiratory failure requiring mechanical ventilation, PVO_2_ had an inverse relationship with arterial-venous oxygen content difference (r = −0.528) and fractional oxygen extraction (r = −0.592). The position of UVC (high in the right atrium or near PFO vs. low in the inferior vena cava) could affect the PVO_2_ measurements and may not accurately reflect the pulmonary arterial PO_2_ especially if they are also being used to infuse fluids.

In our lab, in a meconium aspiration model and a preterm RDS model, a PVO_2_ demonstrated a change point of 25 and 32 mmHg respectively [40,41] (Figure 4). The utility of PVO_2_ from the UVC during the management of PPHN requires further exploration.

### 3.2. Effect of Transfusion, pH and Temperature on Relationship between SpO_2_ and PaO_2_ in PPHN

The influence of transfusion, temperature and acidity (pH) on SpO_2_ and PaO_2_ could be explained by oxygen dissociation curve. Oxygen dissociation curve (ODC) explains the relationship between oxygen saturation of the hemoglobin (plotted in *y*-axis) and oxygen tension (plotted in *x*-axis), which is essential for oxygen absorption in the lungs and delivery to the tissues [42]. A right shift in the curve is associated with release of O_2_ to the tissues and a left shift is associated with O_2_ absorption from the lungs. Fetal hemoglobin (HbF) is the predominant type during fetal and neonatal period. Secondary to high affinity of HbF to O_2_ the ODC is shifted to the left in neonates and this could lead to higher oxygen saturation for lower PaO_2_. Thus, for a PaO_2_ of approximately 40 mmHg, the SpO_2_ could be between 85–93%, and for a SpO_2_ of 97% the PaO_2_ could be > 100 mm Hg [27]. Factors such as blood transfusions affect the ODC. The packed red blood cell transfusion that predominantly has HbA (adult hemoglobin), could alter the oxygen affinity moving the ODC to the right affecting the relation between SpO_2_ and PaO_2_ [42]. PPHN, respiratory failure and carbon dioxide retention with low pH could move the ODC right with lower affinity of Hb to O_2_, which affects the SpO_2_ for a given PaO_2_. Lastly, temperature affects the blood gas analysis unless the values are corrected for body temperature [43].

The solubility of a gas increases with lower temperature and it is important to analyze blood gas and correct it for body temperature. In infants undergoing whole body hypothermia for moderate to severe hypoxic ischemic encephalopathy (HIE), periodic assessment of PaO_2_ corrected for body temperature is important to avoid hypoxia and exacerbation of PPHN. Uncorrected PaO_2_ and SpO_2_ may not be reliable and targeting these values could potentially lead to hypoxemia during whole body hypothermia as shown in Figure 5 [43]. Thus, it is important that blood transfusions, pH and temperature be taken into account while managing an infant with PPHN. Periodically checking arterial blood gases and trying to maintain preductal PaO_2_ in the 50–80 mmHg range may be beneficial in the management of PPHN by avoiding HPV.

## 4. Conclusions

In the management of PPHN, arterial oxygen tension plays an essential role in establishing the diagnosis, assessing the severity, guiding treatment, facilitating specific pulmonary vasodilator therapy, evaluating the response to therapy, and escalating care if needed. Since an UAC placement is standard of care, the disadvantage of having PDPaO_2_ could be a limitation, especially in severe PPHN with labile hypoxemia with ductal shunts. Preductal oxygenation dictates oxygen delivery to brain and heart. Hypoxic pulmonary vasoconstriction is associated with low alveolar oxygen tension and mixed venous oxygen tension. While SpO_2_ provides a continuous, non-invasive assessment of preductal oxygenation, periodic blood gas evaluation is warranted especially in the presence of hypothermia or acidosis. Further clinical trials are warranted to assess the utility of preductal, postductal, and umbilical venous PO_2_ in addition to preductal SpO_2_ in the management of PPHN.

## Figures and Tables

**Figure 1 children-07-00180-f001:**
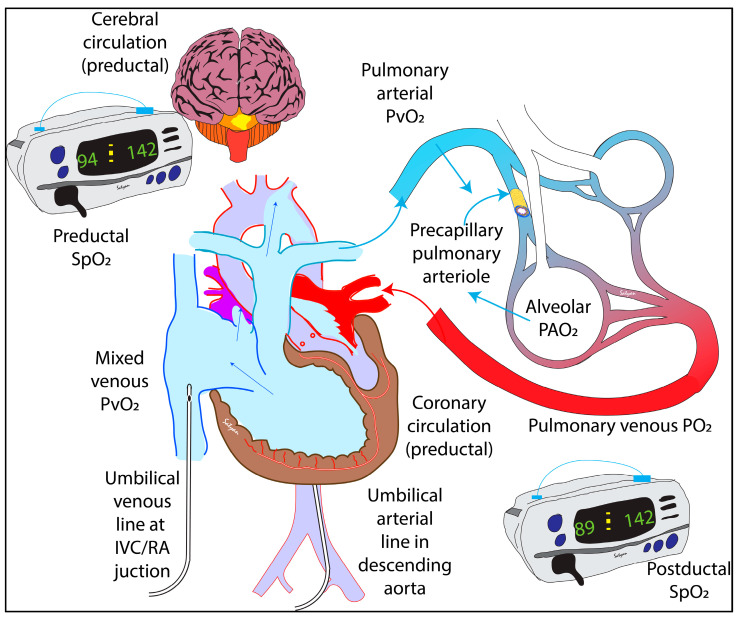
The primary determinant of hypoxic pulmonary vasoconstriction is the precapillary pulmonary arteriole. The oxygen tension at this site is determined by alveolar PAO_2_ and pulmonary arterial PO_2_. In the absence of lung disease, pulmonary venous PO_2_ and alveolar PAO_2_ are usually similar. In the absence of significant shunts, preductal PaO_2_ is reflective of pulmonary venous PO_2_. Umbilical venous PO_2_ is similar to mixed venous PO_2_ and pulmonary arterial PO_2_ in the absence of a left-to-right atrial/ductal shunt. Oxygen delivery to the brain and heart is based on preductal PaO_2_ and SpO_2_. Postductal PaO_2_ (from an umbilical arterial line) can be low and does not predict PVR or oxygen delivery to vital organs. Copyright Satyan Lakshminrusimha.

**Figure 2 children-07-00180-f002:**
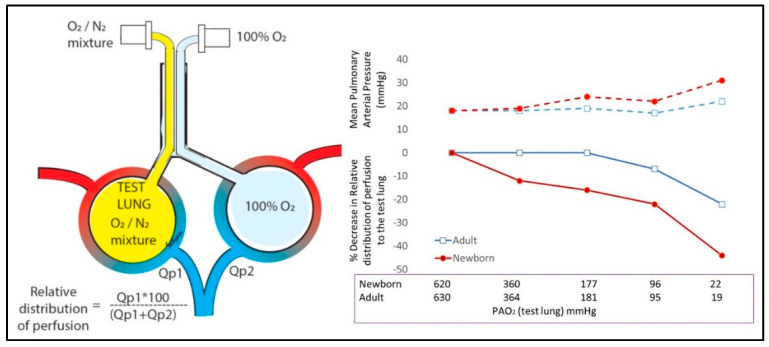
Created using data from Custer et al. [31]. Importance of alveolar hypoxia in directing lung blood flow. Neonatal lambs and adult sheep were instrumented. Each lung was intubated with a different endotracheal tube. One lung was ventilated with 100% oxygen and the other test lung with varying concentrations of oxygen mixed with nitrogen. Relative distribution of perfusion was calculated with Qp1 (blood flow to test lung) and Qp2 (blood flow to the 100% oxygen lung) using the formula shown in the figure. Lower PAO_2_ had a profound effect on pulmonary vasoconstriction in newborn ovine model (red line) compared to adults (blue line).

**Figure 3 children-07-00180-f003:**
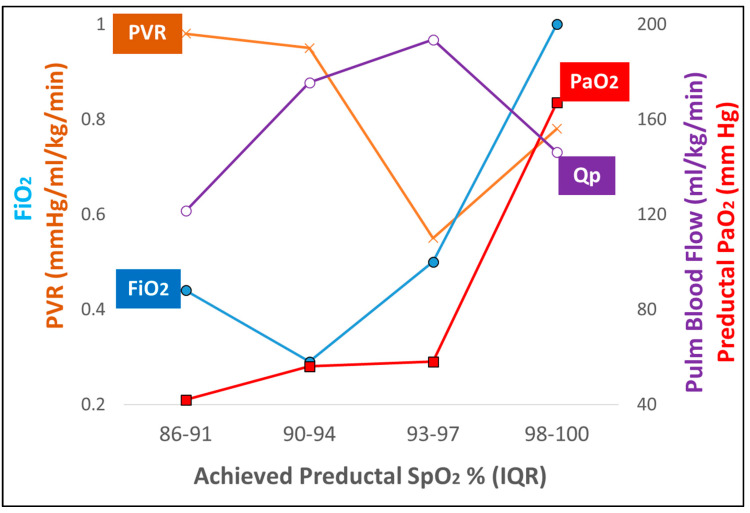
A graph depicting the relationship of inspired oxygen, PVR, left pulmonary blood flow, arterial oxygenation and preductal SpO_2_ is illustrated in an ovine model of meconium aspiration model with PPHN. [36] The pulmonary vascular resistance (PVR—brown cross), pulmonary blood flow (Qp—purple open circles), FiO_2_ (blue circles) and PaO_2_ (red squares) at different preductal saturation (SpO_2_) ranges are shown. Preductal SpO_2_ in high 80 s resulted in high PVR and low Qp. Preductal SpO_2_ in the low−90 s was associated with increased PaO_2_, increased Qp, low FiO_2_ and high PVR (due to high pulmonary arterial pressure–not shown). Preductal SpO_2_ in the mid−90 s was associated with FiO_2_ in the 0.5 range with lowest PVR, highest Qp in this model. Increasing FiO_2_ to 1.0 increased PaO_2_ and SpO_2_ but did not result in further increase in Qp or decrease in PVR. For detailed statistical analysis, please refer to Rawat et al. [36]. Copyright Satyan Lakshminrusimha.

**Figure 4 children-07-00180-f004:**
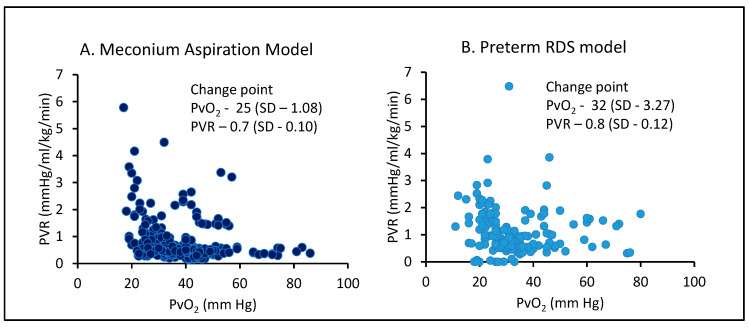
The scatterplot between PVR and PVO_2_ is shown in (**A**) meconium aspiration model and (**B**) preterm RDS model. The PVO_2_ was obtained from the main pulmonary artery blood gas. The MCMC model using SAS 9.4 (NC) estimated change point. Copyright MR/PC/SL.

**Figure 5 children-07-00180-f005:**
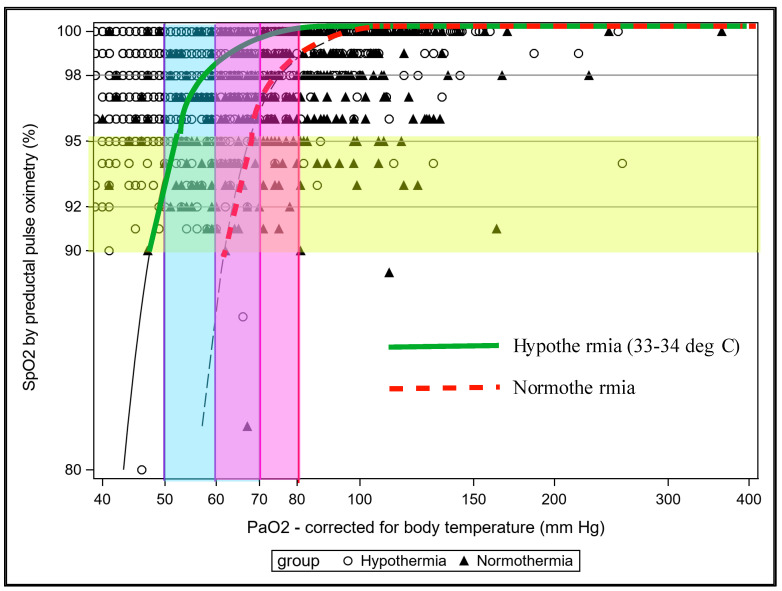
The relation between SpO_2_ (*y*-axis) and PaO_2_ (*x*-axis) in hypothermia and normothermia is shown. In normothermia and hypothermia, the relationship between SpO_2_ and PaO_2_ is altered. Targeting a preductal corrected PaO_2_ of 50–80 mmHg may require preductal SpO_2_ in the mid- to high 90 s during management of severe PPHN on whole body hypothermia. Copyright Satyan Lakshminrusimha.

**Table 1 children-07-00180-t001:** Effect of PAO_2_ on control and PPHN ovine models on PVR [12]

Parameters	O_2_ (%)	PaCO_2_ (mmHg) *	PaO_2_ (mm Hg) *	PVR (mmHg/mL/kg/min) *	PAO_2_ (mmHg) Calculated *
Control	21	42 ± 2	57 ± 6	0.28 ± 0.01	94.5 ± 6.2
PPHN	21	44 ± 3	23 ± 2	1.6 ± 0.2	92.0 ± 5.3
PPHN	50	39 ± 3	36 ± 8	1.0 ± 0.1	301.3 ± 11.3
PPHN	100	47 ± 5	40 ± 5	1.0 ± 0.1	641.3 ± 10.5

* Mean and standard error of mean.

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
