# Peer review of "How Do We Monitor Oxygenation during the Management of PPHN? Alveolar, Arterial, Mixed Venous Oxygen Tension or Peripheral Saturation?"

_children, 2020, doi:10.3390/children7100180_

Round 1

Reviewer 1 Report

This is a well written review article about evaluating the role of alveolar, preductal, postductal, and mixed venous oxygen tension and SpO2 in the management of PPHN. On lines 228-230 you mention “In infants undergoing whole body hypothermia for moderate to severe  hypoxic ischemic encephalopathy (HIE), uncorrected PaO2 may not be reliable and could lead to hypo/hyperoxemia [39] (figure 2).”. The figure 2 you quote is from the reference #39.  This sentence needs to be rephrased for clarity. To avoid confusion you could write “……..uncorrected PaO2 may not be reliable and could lead to hypo/hyperoxemia as shown in Figure 2 in Ref #39” or something similar.

Reviewer 2 Report

This is a review article addressing the role of alveolar, arterial, and venous oxygenation in the care pf PPHN. The authors should be commended on their work in summarizing a very important topic. The manuscript is well written and organized.

In the discussion of iNO, perhaps it would be of interest to address the controversy of using iNO to prevent BPD in preterm babies1 and that the response to iNO is related in part to the stage of alveolar development. In another study by Kinsella et al 2 using lambs to understand the interaction of delivered O2 and iNO in term and preterm babies, and that the response at preterm is such that Qp did not increase. Perhaps this is a particular area in which understanding PAO2 will provide insight.

Some minor edits:

Page 5, line 183: This appears to be the first time that PDPAO2 is mentioned. I couldn’t find an earlier definition of this abbreviation.

Page 5, line 167: Should be “Figure 1”, not “Figure 2”.

Page 6, line 201: Should be “figure 2”, not “figure 1”.

Page 7, line 130: Should this be “figure 1”?

Page 5: Figure 1: The units on the left y column is confusing. It looks like the units are for PVR, but the axis is actually FiO2/PVR, so the unit should be the inverse of PVR.

  1. Lakshminrusimha et al. Just Say No to iNO in Preterms—Really? J of Pediatrics. 2020. V118.
  2. Kinsella et al. Ontogeny of NO activity and response to inhaled NO in the developing ovine pulmonary circulation. Am J Physiol 1994;267:H1955-61.
